# Clinical and Imaging Follow-Up for High-Risk Cutaneous Melanoma: Current Evidence and Guidelines

**DOI:** 10.3390/cancers16142572

**Published:** 2024-07-18

**Authors:** John T. Vetto

**Affiliations:** 1Department of Surgery, Division of Surgical Oncology, Oregon Health & Science University, Portland, OR 97239, USA; vettoj@ohsu.edu; Tel.: +1-503-494-5501; 2Department of Dermatology, Oregon Health & Science University, Portland, OR 97239, USA

**Keywords:** melanoma, high risk, surveillance, imaging, tumor burden, relapse, effective systemic therapies

## Abstract

**Simple Summary:**

Imaging surveillance of patients with high-risk melanoma makes intuitive sense, but supporting data are limited. As a result, guidelines are variable. Future improvements in imaging and the arrival of effective treatments for high-risk melanoma will likely clarify and expand the role of imaging in such patients.

**Abstract:**

The most recent (eighth) edition of the American Joint Committee on Cancer (AJCC) staging system divides invasive cutaneous melanoma into two broad groups: “low-risk” (stage IA–IIA) and “high-risk” (stage IIB–IV). While surveillance imaging for high-risk melanoma patients makes intuitive sense, supporting data are limited in that they are mostly respective and used varying methods, schedules, and endpoints. As a result, there is a lack of uniformity across different dermatologic and oncologic organizations regarding recommendations for follow-up, especially regarding imaging. That said, the bulk of retrospective and prospective data support imaging follow-up for high-risk patients. Currently, it seems that either positron emission tomography (PET) or whole-body computerized tomography (CT) are reasonable options for follow-up, with brain magnetic resonance imaging (MRI) preferred for the detection of brain metastases in patients who can undergo it. The current era of effective systemic therapies (ESTs), which can improve disease-free survival (DFS) and overall survival (OS) beyond lead-time bias, has emphasized the role of imaging in detecting various patterns of EST response and treatment relapse, as well as the importance of radiologic tumor burden.

## 1. Introduction

The most recent (eighth) edition of the American Joint Committee on Cancer (AJCC) staging system divides invasive cutaneous melanoma into two broad groups: “low-risk” (stage IA–IIA) and “high-risk” (stage IIB–IV). Within the high-risk group, the system further distinguishes two subcategories based on nodal status: “high-risk node-negative” (stages IIB and IIC; melanomas that are sentinel lymph node [SLN] negative but have high-risk primary features [thick Breslow depth and/or ulceration]) and “low-risk node-positive” (stage IIIA; melanomas that do not have high-risk primary features but contain micro-metastases in the SLN; Table 1) [1].

While these broad categories are helpful for determining risk level and treatment of patients, they also underscore some inherent problems with the AJCC system. For example, it is well known that stage IIIA melanoma has a better prognosis than stage IIB–IIC, despite the numbering of the stages. Further, likely due to the fact that 92% of patients are initially diagnosed with stages I and II melanoma (excluding the minority diagnosed as stage IV at initial presentation), the majority (60%) of patients who go on to die of melanoma were stage I and II at initial diagnosis. This latter fact is particularly important when one considers the most recent data from the National Cancer Institute (NCI) Surveillance, Epidemiology, and End Results (SEER) program, and as well epidemiological data reported by Olsen et al., which show that the incidence of melanoma, particularly early-stage melanoma, is continuing to rise, especially among populations at highest risk [2]. It is estimated that in 2024 there will be 100,640 new cases of melanoma resulting in 8290 deaths. As a result, cutaneous melanoma now results in 5% of all cancer diagnoses, and 1.4% of all cancer deaths. However, because most patients with cutaneous melanoma will survive after resection alone, the relative 5-year survival of all patients is 94.1% [3], a favorable number that may down-play the importance of follow-up screening in suspectable patients.

It is essential, therefore, when considering the possible benefit of follow-up in melanoma, that an emphasis be placed on patients with sufficient risk of relapse to warrant such follow-up. Accordingly, this review will focus in particular on AJCC high-risk patients and also on newer methods of identifying patients staged as AJCC low-risk who actually have high risk of relapse and death.

Methods: To conduct this review, the author reviewed and compared current major guidelines (Table 1) and also searched Pubmed (pubmed.ncbi.nim.nih.gov) with the search terms “melanoma”, “imaging”, follow-up”, and also “review”. This resulted in the identification of 20 major relevant studies that form the bulk of the review (Table 2).

## 2. Historical Considerations

In a 2013 review of melanoma follow-up guidelines, Trotter and colleagues noted a lack of uniformity across different dermatologic and oncologic organizations. Reviewing National Comprehensive Cancer Network (NCCN) [4], European Society for Medical Oncology (ESMO), American Academy of Dermatology (AAD), and British Association of Dermatologists (BAD) guidelines, and guidelines from Germany, Switzerland, and Australia/New Zealand for follow-up of high-risk melanoma (AJCC stage IIB–IV), they found that while regular physical examinations were recommended by six of the seven groups (two recommending that exams be carried out indefinitely), there was no consensus for testing [5]. Specifically, two groups recommended no testing, two recommended S-100beta (a marker popular at the time that has since been associated with high false positive and false negative results [6]) and consideration of scans for 5 years, one recommended regular scanning but with no details (ESMO), and two recommended that imaging be “considered”. Of these latter two, one was the NCCN, which at the time suggested consideration of chest X-rays or CT scans every 3 to 12 months and an annual brain MRI for 5 years [5].

In the 11 years that have passed since that review, dramatic changes have occurred in the treatment of high-risk melanoma that improved the prognosis, and call for a re-examination of follow-up guidelines for such patients. These changes have included, but are limited to, the advent of the “Era of Effective Systemic Therapies” (EST; immunotherapy and targeted therapies [7]), the elimination of completion lymph node dissection as routine treatment for positive sentinel lymph node(s) [8], and the approval of adjuvant immunotherapy for high-risk node-negative patients (stage IIB–IIC) [9].

## 3. Recent General Reviews

Regarding initial work-up, a 2022 collaboration of multi-disciplinary experts from the European Dermatology Forum (EDF), the European Association of Dermato-Oncology (EADO) and the European Organization for Research and Treatment of Cancer (EORTC) was formed to make recommendations on CM diagnosis and treatment, based on systematic literature reviews and the experts’ experience. The panel found that thin melanomas up to 0.8 mm in tumor thickness do not require any imaging diagnostics. From stage IB onwards, examinations with lymph node sonography were recommended, but not further imaging examinations. From stage IIC onwards, initial whole-body cross-sectional radiologic examinations with computed tomography (CT) or positron emission tomography CT (PET-CT) in combination with brain magnetic resonance imaging (MRI) were recommended. From stage III and higher, mutation testing (specific testing for actionable mutations in BRAF, NRAS, c-kit, etc., or multi-gene testing) was recommended, particularly for the BRAF V600 mutation.

The panel concluded that multidisciplinary management in melanoma is key and emphasized the importance of providing structured follow-up to detect relapses and secondary primary melanomas as early as possible [10]. This recommendation was consistent with the concept of identifying relapses at the lowest possible disease burden (see “Rationale for Early Detection of Relapses”) [11]. However, regarding follow-up, the panel also concluded that there is no evidence to define the frequency and extent of examinations.

Regarding follow-up, a recent Scoping review of 26 articles from 19 different organizations found that life-long annual skin surveillance with a physician was recommended by 53% (9/17) of guidelines. Routine laboratory investigations were recommended by 7/19 guidelines. Regional lymph node ultrasound was recommended by 9/16 guidelines, most often in stage IB or higher, and was optional in 7/16 for patients who met specific criteria. Surveillance with PET-CT or CT and MRI was recommended by 15 and 11 guidelines, respectively, most commonly in stage IIC or higher, with a variable frequency and total duration. Five out of nine guidelines indicated a preference for skin surveillance to be completed with a dermatologist [12]. Similar to the 2013 review, the authors concluded that guidelines remain highly variable for many aspects of melanoma surveillance, which may be partly attributed to regional differences in healthcare workforce distribution and availability of imaging technologies [5,12].

## 4. Imaging in the Era of EST: Special Considerations

ESTs, such as BRAF-targeted therapies and immunotherapies [7], have brought new importance to imaging in melanoma. In older studies and guidelines, it was difficult to show an advantage to imaging for follow-up because of the lack of effective treatments when progression was detected. Any potential benefit of earlier detection of metastatic progression was often written off to “lead-time bias”. Lead-time bias, a bias specific to screening studies, occurs when a disease is detected by a screening or surveillance test at an earlier time point than it would have been if it had been diagnosed by symptoms or examination findings (clinical detection). If survival time is measured from the time of cancer diagnosis, then the apparent increased survival time in the screened as compared with the control group is an artifact; a screened and unscreened patient may actually have the same survival but the screened patient will appear to have lived longer because of the earlier detection. Lead-time bias has been used for decades and in several types of cancer as an argument against screening studies, especially when there are no effective treatments for the recurrences once they are detected [13].

However, with the advent of ESTs in melanoma that can improve overall and melanoma-specific survival of detected progressions (e.g., the improvements in survival in advanced melanoma seen in the CHECKMATE 067 trial comparing dual-agent to single-agent immunotherapy [14]), imaging has the previously mentioned advantage of detecting progression at a lower disease volume (see also “Rationale for Earlier Detection of Relapses”) [11]. In addition, the use of EST has resulted in new patterns of relapses (see below) [15] and the ongoing need to assess responses to therapies, especially because we now have multiple ESTs with different mechanisms of action, which allow for switching from one treatment to another [7].

With the exception of nodal ultrasound, most older and “lower technology” screening tools, such as chest X-ray and blood tests (e.g., LDH, S-100B), while relatively inexpensive, have low sensitivity and have been proven ineffective as a screening and follow-up tool in U.S. and European studies [6]. In contrast, cross-sectional imaging, namely CT and PET-CT, provide high-resolution anatomic information and are recommended at baseline for stage III–IV by the NCCN, NICE, German, and Australian/New Zealand guidelines [16,17]. These guidelines generally recommend either CT or PET-CT; a 2011 met-analysis by Xing et al. found that PET-CT had better sensitivity and specificity for detecting metastases (80% and 87%, respectively) compared to CT (51% and 69%, respectively) [18]. However, the use of PET-CT is limited by the delivery of higher energy positron radiation, expense, and availability, and recent years have seen dramatic improvements in the sensitivity of whole-body CT. A 2024 study showed no difference in sensitivity and specificity between PET-CT and whole-body CT when used to stage advanced ovarian cancer [19]. Peritoneal implants can have low uptake on PET-CT; correlation with CT and PET-CT is essential when such lesions are suspected.
cancers-16-02572-t001_Table 1Table 1AJCC melanoma risk categories1 and corresponding NCCN recommendations for imaging [4].CategoryAJCC StagesNCCN Recommendations for Imaging

BaselineSurveillance *“Low-risk”Stage IA–IIANoneNone“High-risk nodenegative”Stages IIB and IICCx-sectional imagingTo investigate

w/w-o Brain imaging specific signs and

“if needed for surgical symptoms; every planning” or prior to3–12 months for

starting adjuvant Rx2 years, then every 6–12 months for another 3 years; not recommended after 3–5 years +“Low-risk node-positive”Stage IIIAConsider cx-sectionalAs above

imaging
“Advanced/Metastatic”Stage IIIB–IVPerform cx-sectionalAs above

imaging and brain


imaging
AJCC = American Joint Committee on Cancer; NCCN = National Comprehensive Cancer Network; cx = cross; w/w-o = with or without; * if no evidence of disease (NED); + depending on risk of relapse.


Although it is well known that melanoma can metastasize to multiple sites, screening/follow-up imaging carried out for melanoma should specifically be able to detect metastases to bone, GI tract, liver, lung, and brain, as well as nodes and soft tissue. Bone scintigraphy is not often conducted in melanoma patients, and bone metastases—which are infrequent in melanoma—are best found using PET. Liver metastases are rarely the first metastatic site in melanoma but are found in 10–12% of patients with stage IV disease. They can be detected by either CT or PET, but when suspected on those studies their presence is best confirmed by MRI in patients who do not have a contraindication to magnetic imaging [15]. GI metastases from melanoma are the most common cause of malignant bowel obstruction in adults, and a 2016 study showed that PET-CT was more sensitive and specific than CT for the detection of such lesions [20]. Conversely, lung CT generally has greater sensitivity than PET for the detection of smaller pulmonary metastases; a study by Reinhardt et al. showed that even with the addition of a low-dose attenuation-corrected CT, PET-CT often cannot detect lung nodules less than 5 mm in size [21]. Taken together, these considerations support the fact that most guidelines for imaging of high-risk melanoma list both CT and PET as options [16,17,22]. Regarding PET-CT, it is particularly essential that international standardized procedures for this study be followed to assume uniform results between centers.

As detailed in a review by Stodell, the use of these imaging modalities to follow melanoma patients treated with ESTs has led to the discovery of unique radiologic response patterns, including periods of stability of index lesions followed by shrinkage, heterogenous (“mixed”) responses, the persistence of one or two lesions (oligometastases, which can then be referred for metastasectomy [23]), and pseudoprogression [15]. This latter phenomenon results from inflammation and T cell infiltration of index lesions early in treatment, which can give the appearance that lesions are growing. The possibility that enlargement of a lesion on imaging is pseudoprogression (and not actual progression) can be confirmed by obtaining a PET-CT (which can show a decrease in the SUV of the lesions) or simply continuing treatment and observing shrinkage on subsequent scans. These phenomena have led to the development of a revision of the standard RECIST (response evaluation criteria in solid tumors) criteria for treatment response used by radiologists. This revision, known as the immune-related response criteria (irCR), re-define responses by looking at changes between serial scans [24].

## 5. Follow-Up for Patients with High-Risk Stages (Table 2)

For resected AJCC high-risk melanoma, several studies show that first recurrences are most commonly systemic, and therefore cross-sectional imaging studies should be carried out as part of follow-up. Currently, the NCCN recommends imaging follow-up for high-risk resected melanoma (IIB-IV) with either PET or CT scan for the first 5 years after completion of treatment, although the interval for imaging in the NCCN is listed as “every 3 to 12 months” and is left up to the provider. Since most recurrences in high-risk patients occur within the first 2 to 3 years, the NCCN does not recommend imaging after 5 years (Table 1) [4,16]. Most providers pick an interval of every 4 to 6 months for 5 years. As already mentioned, the NCCN also does not specify whether the imaging should be CT or PET-CT [4]; while CT scans are generally less expensive and use X-rays rather than positron radiation, PET has the advantage of imaging the extremity without the need for a separate scan.

## 6. Retrospective Studies

A 2017 study from Ohio State University by Kurtz et al. looked at 247 patients with stage II–III melanoma treated with operation from 2009–2015 who were followed for a minimum of 6 months after surgery but without a standardized follow-up schedule. Since the seventh edition of the AJCC staging system was in use at the time of the study, the stage groupings the authors used were not relevant to the current staging group definitions noted in the Introduction. Regardless, grouping stages IIA and IIB together (patients with T2b-T4a tumors and negative nodes), they found a recurrence rate of 11% (14/125); 80% (11) of these were found by history or physical examination. About two-thirds of these patients received at least one chest X-ray (beginning at an average time of 4.7 months post-op) while the rest had no imaging. At a median follow-up of 35 months there was no difference in survival between those patients who had the chest X-rays and those who had no imaging [95% CI: 0.88–0.99]).

However, for stage IIC/IIIA–C patients (node-negative T4b tumors or any positive nodes), 87% of patients had at least two cross-sectional imaging studies (whole-body CT or PET-CT) plus brain MRI. Twenty-three percent (28/122) of patients recurred at a median follow-up of 31 months. Fifty percent of recurrences were detected by imaging in asymptomatic patients, whereas the other 50% (14/28) were detected clinically and confirmed on imaging. Of the 28 stage IIC/IIIA–C patients who recurred, 57% (16/28) had surgical resection of the recurrence, whereas 11 (39%) patients received up-front systemic therapy.

The authors concluded that there was no role for chest X-rays in stage IIA–IIB melanomas, whereas stage IIC–IIIA–C melanomas were associated with higher recurrence rates and routine whole-body imaging seemed more useful for detecting recurrences and triggering additional surgery and/or systemic therapy [25]. In addition to the issue of the use of the earlier AJCC staging system, this study pre-dated the expanded use of adjuvant therapy, especially for high-risk node-negative patients.

In a 2018 study from the U.K. by Lim et al. which updated staging to the eighth edition of the AJCC staging system, 173 patients, mostly (79%) with treated stages IIIB and IIIC of disease, were followed for a median of 23.3 months with a protocol consisting of a surveillance schedule of CT or PET-CT and brain MRI every 6 months for 3 years, then annually in years 4 and 5, as well as clinical examination conducted at the same intervals. This protocol was based on a prior U.K. consensus conference recommendation, and the outcomes of patients who had been followed on it were evaluated retrospectively.

During the study period, 82 patients (47%) relapsed with a median time to relapse of 10 months and a median relapse-free survival (RFS) of 21 months. Most recurrences (66%) were asymptomatic and detected by the scheduled surveillance scans. Sixty-eight percent of patients recurred with stage IV disease, with a median overall survival (OS) of 25.3 months; the remaining 32% of patients had an initial loco-regional recurrence, with a median OS not reached (*p* = 0.016). Patients who underwent surgery at recurrence for either stage III (27%) or IV (18%) disease did not reach their median OS. The median OS for the 33 patients (40%) who received systemic therapy was 13 months.

The authors concluded that imaging appeared to reliably detect subclinical disease and identify patients suitable for surgery, resulting in favorable outcomes. The authors also felt that the short median time to relapse in their study supported the idea of intensifying imaging scheduling in the first year of surveillance [26]. This latter conclusion agreed with an older retrospective Australian study by Francken et al. of 873 patients, which found that most relapses occurred early in the first few years after surgery (even though their patients were AJCC stages I and II) [27]. These and other studies reveal that despite early-stage melanoma’s reputation for late recurrence, most melanoma relapses occur earlier, in keeping with the recurrence patterns of other solid malignancies. This underscores the potential utility of image follow-up, especially for high-risk melanoma.
cancers-16-02572-t002_Table 2Table 2Summary of recent (2016–present) studies on surveillance imaging of high-risk melanoma.YearAuthorCountrynStagesBenefit of Imaging?*Retrospective*2017Kurtz [25]U.S.247IIA–IIB *No



IIC–IIIA–C *Yes2018Lim [26]U.K.173IIIB–IIICYes2018Lewin [28]Australia170All stage IIIYes +2020Ibrahim [29]Canada353IIIB–IIICYes2023Yan [30]Australia199IIC–IIIYes*Prospective*2016Podlipnik [31]Spain290IIB–IIIYes2020–PresentTRIM Study [32]Sweden1300IIB–IIIPendingn = number of patients; U.S. = United States; U.K. = United Kingdom; * used 7th edition AJCC; + examined only PET-CT.


Further, the poorer OS of patients treated with systemic therapy in this study probably reflected the relatively inferior treatment options during the study period (2013–2015), as this was relatively early in the era of EST. However, a recent report of metastasectomies performed for stages III and IV disease during the era of EST also found a benefit of resection, similar to the Lim study [26], which the authors attributed to earlier detection with resultant smaller disease burden and fewer metastatic deposits [23].

In a study from Australia published the same year as the Lim study (2018), 170 patients from a PET-CT database were identified who had regularly undergone routine PET-CT, with or without brain MRI, on a schedule similar to the U.K. Relapses were detected in 38% of patients, most of whom (69%) were asymptomatic. At a mean of 49 months, 33 (52%) of the patients underwent potentially curative metastasectomy, and 10 remained disease-free after a median follow-up of 24 months. The authors concluded that using PET-CT for follow-up of stage III melanoma allowed for detection of most recurrences while still asymptomatic and was associated with a high rate of detection of resectable and potentially curable disease. Interestingly, these authors also noted that PET-CT in this setting had a high NPV (89–96%) that could provide patient reassurance [28]. This idea that a negative scan provides patient benefit is apparent to most clinicians but runs counter to economic arguments that negative scans produce medical “harm”.

A 2020 retrospective study by Ibrahim et al. from the University of Ottawa showed that follow-up with imaging in high-risk patients led to earlier detection of recurrences and better survival, presumably because patients were referred sooner for systemic disease, with lower tumor burdens. Specifically, these workers followed 353 high-risk but non-metastatic melanoma patients (stage IIB–IIIC) with imaging (which was not the standard of care in Canada at the time). One hundred fifty-nine (45%) recurred during the study period, similar to the study by Lim [26]. Of those, the authors compared the 71 (45%) who had had asymptomatic study-detected recurrence (ASDR; i.e., recurrences noted only on imaging) to the 88 (55%) who presented with symptoms or signs of recurrence (symptomatic recurrence [SR]). Not surprisingly, shorter imaging intervals identified more ASDRs (57%, 0–6 months; 34%, 6–12 months; 33%, >12 months; *p* = 0.03), and ASDR was associated with better prognostic factors than SR; ASDR patients were more likely to have fewer than three metastatic sites (43 vs. 21%, *p* = 0.003), normal lactate dehydrogenase (LDH; 53 vs. 38%, *p* = 0.09), and less often had associated brain metastases (11 vs. 40%, *p* < 0.001)].

In terms of outcomes, ASDR patients received more systemic treatment (72 vs. 49%, *p* = 0.003) and had better survival outcomes on immunotherapy (2-year OS, 56 vs. 31%, *p* < 0.001). The median OS from surveillance start was 39.6 months for ASDR patients vs. 22.8 months for SR patients. (*p* < 0.001). Most importantly, ASDR was independently associated with survival (*p* = 0.003), adjusting for stage, sex, age, disease burden, LDH, era of recurrence, brain metastases, and type of systemic treatment [29]. In a similar study from Melbourne Australia, Yan and colleagues followed 199 patients with high-risk (stage IIC–III) patients with PET-CT. Fifty-three presented recurrences that were asymptomatic and detected first on scanning (i.e., ASDRs). The authors found that ASDRs were associated with better survival compared to SRs (hazard ratio 0.25, 95% CI 0.10–0.66, *p* = 0.05) [30]. Both studies concluded that an intensified imaging surveillance protocol for high-risk resected melanoma was associated with superior survival outcomes and should be studied prospectively [29,30].

## 7. Prospective Studies

Prospective studies of scanning in high-risk melanoma patients are few. In a large German study from 2003, Garbe et al. followed consecutive patients with stage I–IV melanoma by the then German guidelines for a 3-year period (1996 to 1998) and reported 5-year actual data. The 2008 patients enrolled underwent 3800 clinical examinations and 12,398 imaging exams (thus, the guidelines stressed imaging over physical examination). There were 233 recurrences in 112 patients. In stage I–III disease, half of the recurrences were detected by physical examination, and 21% were discovered by nodal ultrasound. Thus, most recurrences were loco-regional, and these patients did significantly better compared to historical controls [33]. The application of these results to the modern era of EST is marred by several factors, including the use of the seventh edition AJCC staging system, the use of nodal ultrasound and chest X-rays instead of cross-sectional imaging (which likely skewed the recurrence detection toward loco-regional disease), the underuse of modern dermatology follow-up, and the lack of EST.

In a more modern prospective study performed in Spain, almost half of the 115 relapses among 290 patients were detected by imaging. Specifically, this study prospectively followed a cohort of 290 consecutive AJCC high-risk patients (stages IIB, IIC, and III) with an intensive imaging protocol (whole-body CTs, and brain MRIs), along with periodic laboratory tests, regular physical examinations, and patient self-examinations. A total of 2382 clinical examinations and 3069 imaging tests were performed. At a median of 2.5 years of follow-up there were 115 recurrences, of which the method of first detection, in order, were whole-body CT (48.3%), provider history and physical (23.7%), patient self-reporting or self-examination (17.8%), brain MRI (7.6%), and lastly, laboratory tests (2.5%).

Although the study was marred by the grouping of the relatively small number of stage III patients together and the lack of actual survival data, it provided support for image follow-up of AJCC high-risk node-negative patients (stage IIB, IIC) that pre-dated the current FDA approval of adjuvant immunotherapy for this subgroup. The authors concluded that intensive monitoring was appropriate for early detection of recurrence in stage IIB, IIC, and III melanomas [31].

In 2020, Sweden launched TRIM, an ambitious, prospective, randomized, 20 institution, national study to compare their national guidelines for melanoma screening to a more intense follow-up regimen. In distinction from other Nordic countries where PET-CT is used routinely to follow high-risk patients for up to 5 years, in Sweden the guidelines state that routine blood tests and imaging should not be used; patients are followed with clinical examinations only, and only for 3 years after diagnosis of stage I–III melanomas. The exception is patients with a positive sentinel lymph node (SLN) biopsy who do not have a completion node dissection; these patients receive nodal ultrasounds.

In the TRIM study, patients with high-risk non-metastatic melanoma (IIB–III) were randomized to follow-up with either the rather conservative Swedish national guidelines described above, or a more intense regimen of the clinical examination recommended by the guidelines plus routine CTs or PET-CTs and blood testing with S100B and LDH. Patients with limited life expectancy or who would not be offered systemic therapy were excluded. Patients eligible for adjuvant therapy (i.e., stage III, at the time the study was written) were included. The first patient was enrolled in June of 2017, and enrollment completed in 2021 with approximately 650 patients in each arm. All patients will be followed for 5 years, with a primary endpoint of overall survival (OS) and secondary endpoint of disease-free survival. The study will include extensive quality-of-life and economic analyses [34].

Although the final results of this trial are pending, and preliminary results (unpublished) suggest no difference in OS, the fact that both arms received adjuvant therapy for stage III disease and nodal ultrasounds for undissected nodal beds with a positive SLN (nodal progression is the most common site of first progression in melanoma) may prevent the study from showing a survival advantage to imaging. Further, the facts that most medical oncologists favor imaging for patient follow-up on adjuvant therapy and that an increasing number of countries are approving adjuvant therapy for high-risk node-negative patients may render the findings somewhat moot.

## 8. Early Detection of Relapses in AJCC Low-Risk Patients

As discussed previously, several studies show that the main recurrence risk of American Joint Committee on Cancer (AJCC) “low-risk” melanoma stages (IA–IIA) are loco-regional, with new primaries, local recurrences, in-transit lesions, and nodal progression [3,10,12,33]. Most of these events are detected by patient or provider and not by imaging. Therefore, the NCCN recommends only skin and nodal exams for such patients, which can often be conducted with dermatology-only follow-up visits [4,5].

In fact, historically, there has been little interest in scanning for AJCC low-risk patients, and that sentiment is supported by prospective data. For example, the MELFO (MELanoma FOllow-up) study was an international phase III randomized controlled trial conducted in the U.K. and the Netherlands, which compared low-intensity follow-up to a higher-intensity regimen. Conducted between 2006 and 2016, the study enrolled 388 adult patients (196 high-intensity, 192 low-intensity with sentinel node-negative primary melanoma who were mostly AJCC low-risk stages). At 5 years of actual follow-up (the study was reported in 2022), both arms expressed high satisfaction (>97%) with their follow-up schedules. In total, 75/388 (19.4%) patients recurred, with no difference in incidence found between the two arms (*p* = 0.57). Among those patients, the major method of recurrence detection was self-examination (25 experimental and 32 control patients; 75.8% vs. 76.2%; *p* = 0.41). The study found no difference in any survival outcomes between the two study arms (*p* = 0.99 for DFS). The authors concluded that a reduced-intensity schedule, especially for AJCC low-risk stage patients, was patient-friendly and safe, and that patient self-examination was effective for recurrence detection with no diagnostic delay [32].

However, there were two notable features in the study that may not by apparent to U.S. readers. First, the study used the U.K.’s NICE (National Institute for Health and Care Excellence) guidelines; even for the low-impact regimen, these include nurse-led and physician-supervised full histories and total body physical (not just skin) examinations, as well as nodal bed ultrasounds, for 5 years [17]. Regarding nodal bed ultrasounds, while ultrasounds were recommended by the MSLT-2 study for patients with node-positive melanoma who did not undergo completion node dissection [8], that study predated current adjuvant therapy and NCCN guidelines, which obviates the need for ultrasounds. Further, the MSLT-1 study demonstrated that the average time for recurrence in an undissected nodal bed was 19 months [35]. Thus, following the node basins out to 5 years with the U.S. is unnecessary. Even the original MSLT-2 study criteria monitored patients with ultrasound only every 4 months during the first 2 years, and then every 6 months during years 3 through 5. In that study, most nodal recurrences occurred by 2 years [8]. Thus, the NICE guidelines used in the MELFO were excessive by U.S. standards and go well beyond NCCN recommendations [4].

Secondly, 19% of patients in each arm were actually AJCC high-risk, and accounted for most of the relapses; currently in the U.S., such patients would be offered systemic immunotherapy followed with imaging. Thus, the MELFO study did not really focus on NCCN follow-up guidelines in AJCC low-risk patients.

More recently, and with the extension of adjuvant systemic therapy to AJCC high-risk node-negative patients (stage IIB–IIC) resulting from the KEYNOTE-716 and CHECKMATE -76K trials [9], interest is focusing on further extending such treatment to AJCC low-risk patients who are actually at high risk for progression biologically. It is well known that some patients classified as “low-risk” via the AJCC staging system will suffer systemic progression [1]. As previously mentioned, 92% of patients are initially diagnosed with stages I and II melanoma (excluding the minority diagnosed as stage IV at initial presentation), yet the majority (60%) of patients who go on to die of melanoma were stage I and II at initial diagnosis [2]. While high-risk node-negative patients (stage IIB–IIC) account for some of these deaths, they do not account for all, meaning that some patients with AJCC low-risk disease (stages IA–IIA) will progress and die, despite being node negative.

Thus, although most AJCC low-risk patients would not benefit from imaging, those who are actually biologically at higher risk might. Multiple studies (including the previously discussed prospective study from Spain [31]) have shown that the majority (75–96%) of first recurrences in melanoma are detected by imaging, with patients and providers finding only the remaining 4–25% [36,37]. As early as 2010, Lieter and colleagues in Germany reported that this earlier detection of metastases in stage I–II could lead to improved survival over that expected by lead-time bias [38]. That observation was made before the arrival of ESTs, which, as already discussed, can improve MSS and OS, and are most effective against lower tumor burdens [11]. This correlation between lower tumor burden and best treatment responses has been shown for both targeted therapy [39] and immunotherapy [40]. Further, data such as the previously discussed retrospective studies by Ibrahim [29] and Yan [30], show better survival when recurrences are detected on imaging and are still asymptomatic, versus waiting for symptoms to appear.

Taken together, all these considerations suggest a future strategy for improving survival in Stage IA–IIB “low-risk” patients:Identify patients with low-stage tumors that are actually at biologically higher risk for relapse;Target imaging to those patients with the goal to detecting relapse at lower tumor volume;Treat those patients earlier with ESTs.

## 9. Using Genetic Expression Profiling to Re-Define “High Risk”

Of the three goals outlined above, the most difficult is the first: finding AJCC low-risk patients (stage IA–IIB) who are actually biologically at higher risk for relapse. One method that is being actively investigated is genetic expression profiling using grouped signatures of genes that predict for risk of relapse called genetic expression profiles (GEPs). A 31 gene signature (31-GEP, Decision Dx Melanoma, Castle Biosciences, Friendswood, TX) has been shown to identify node-negative patients with a higher risk of relapse [41]. In a cohort study of 393 pathologically staged (i.e., node-negative) low-risk melanomas, 46 (12%) had the highest genetic risk score for relapse (“Class 2B”). These Class 2B patients had a 76% 5-year distant metastasis-free survival, compared to 97% for the lowest genetic risk group (“Class 1A”; *p* < 0.0001) [42]. Thus, genetic tests can identify a small but real group of AJCC low-risk patients who could benefit from imaging follow-up. GEPs look beyond standard clinical and histopathologic features of the tumor to define “high risk” more broadly.

To explore this possibility, Dhillon and colleagues prospectively followed a cohort of 307 sentinel node-negative patients from three institutions who had higher-risk GEPs (Classes 2A and 2B) with serial whole-body CT scans conducted every 6 months until there was evidence of progression (“experimental group”). The investigators compared the outcomes of the experimental group to 327 patients treated at the same institutions who had not had the GEP and were not being scanned. Using a modified RECIST method to determine total tumor volume [43], they found that at first recurrence the scans from the experimental group had significantly less average tumor burden than the control group (27.6 mm^2^ vs. 73.1 mm^2^). Most importantly, more of the experimental patients than the controls were able to start subsequent immunotherapy (76.3% vs. 67.9%), and at 45.6 months of mean follow-up the OS was significantly better for the experimental group (76% vs. 50% [*p* = 0.027]), even though the experimental patients had higher mean age and thicker primary Breslow levels [44]. The improvement in survival from earlier tumor detection was similar to that reported by Ibrahim et al. (see “Follow-up for Patients with High-Risk Stages/Prospective Studies”) for AJCC high-risk patients [29], supporting the concept that “high risk” can be defined by genetic as well as histopathologic factors. The Dhillon et al. study was limited by selection bias, the inclusion of some AJCC high-risk node-negative patients, and retrospective analysis [44], and therefore a prospective randomized trial is planned.

## 10. Conclusions

The most recent (eighth) edition of the American Joint Committee on Cancer (AJCC) staging system divides invasive cutaneous melanoma into two broad groups: “low-risk” (stage IA–IIA) and “high-risk” (stage IIB–IV) [1]. While these broad categories are helpful for determining risk level and treatment of patients, they also underscore some inherent problems with the AJCC system. Furthermore, because the bulk of patients with newly diagnosed melanomas are in the “low-risk” category, it is essential, when considering the possible benefit of follow-up, that emphasis be placed on patients with sufficient risk of relapse to warrant such follow-up. Unfortunately, there continues to be a lack of uniformity across different dermatologic and oncologic organizations regarding recommendations for follow-up, especially in regard to imaging [5]. That said, the bulk of retrospective and prospective data support imaging follow-up for high-risk patients.

Screening/follow-up imaging carried out for melanoma should specifically have the ability to detect metastases to bone, GI tract, liver, lung, and brain, as well as nodes and soft tissue [15]. Currently, it seems that either PET or whole-body CT are reasonable options for follow-up, with brain MRI preferred for the detection of brain metastases in patients who can undergo MRI [16,17,22]. The current era of ESTs, which can improve DFS and OS beyond lead-time bias [14], has revealed the importance of imaging in detecting various patterns of EST response and treatment relapse, and also emphasized the importance of radiologic tumor burden [11]. Future research will take into account tumor burden at relapse rather than just stage-related risk of relapse.

## 11. Future Directions

Referring to the AJCC categories shown in Table 1, the author believes that the near future will bring exciting changes in imaging for melanoma. As discussed in the previous section, for AJCC low-risk melanomas (stage IA–IIA) better risk stratification (e.g., with GEP) will identify those patients at higher risk for progression, independent of stage, who might benefit from image surveillance [44]. For high-risk node-negative (stage IIB–IIC) and low-risk node-positive patients who are currently receiving adjuvant therapy but for the lowest potential benefit, the fact that these patients are also more often being imaged will allow the amassing of a large amount of data that will at least have the potential for large retrospective analysis on the benefits/downsides of imaging follow-up and improve the quality of level 2 evidence.

For high-risk node-positive and metastatic patients (stage IIIB–IV), the continued study and refinement of circulating tumor DNA (ctDNA) assays may lead to more selective (and therefore more cost-effective and informative) use of scanning—that is, the ordering of scans only in patients with rising ctDNA titers. In 2021, a met-analysis of 19 studies revealed that in most studies rising ctDNA titers was a negative prognostic factor that preceded the finding of progression on imaging. The authors called for better standardization of ctDNA techniques [45]. A 2024 review of 15 papers (6 in locally advanced melanoma and 9 in metastatic melanoma) also concluded that there was ongoing need to better define precise and dynamic biomarkers for ctDNA assays [46].

Finally, for all melanoma stages, the ongoing radiologic science trends of developing newer generations of imaging will continue, producing imaging techniques that are less expensive, safer, more widely available and even portable, more linked (allowing for better inter-scan interpretations, including interpretation with artificial intelligence), and have increased sensitivity and specificity [47]. Further research and experience in the era of ESTs will result in a better understanding of the utility and pitfalls of imaging studies (especially PET-CT) obtained during adjuvant, neoadjuvant, and palliative systemic treatments [48].

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
