# Peer review of "Clinical and Imaging Follow-Up for High-Risk Cutaneous Melanoma: Current Evidence and Guidelines"

_cancers, 2024, doi:10.3390/cancers16142572_

Round 1

Reviewer 1 Report

Comments and Suggestions for Authors

I have read John T. Vetto's paper "Clinical and Imaging Follow-up for High-Risk Cutaneous Melanoma: Current Evidence and Guidelines" with interest.

Most retrospective and prospective data supports imaging follow-up for high-risk melanoma patients. At present, either positron emission tomography (PET) or whole-body computerized tomography (CT) are reasonable options for follow-up, with brain magnetic resonance imaging (MRI) preferred for the detection of brain metastases in patients who can undergo it. The current era of effective systemic therapies that can improve disease-free survival (DFS) and overall survival (OS) beyond lead-time bias has emphasized the role of imaging in detecting various patterns of treatment response and relapse and the importance of radiologic tumor burden. 

Also, the review of studies focused on low-risk melanoma patients was comprehensive, and the future direction that the Author pointed out here is applicable.

Minor observation: table 2 must be improved because it is not easily understandable.

Author Response

Minor observation: table 2 must be improved because it is not easily understandable.

The table contained formatting errors. These have been fixed.

Reviewer 2 Report

Comments and Suggestions for Authors

The author submitted a systematic review investigating the role of clinical and imaging follow-up in cutaneous melanoma. The author reports on a relevant topic since there increasing incidence of melanoma and different follow-up regimes in different countries with need for a systematic investigation of these strategies to gain evidence-based treatment decisions.

The introduction provides sufficient background information on cutaneous melanoma, the current imaging follow-up regime and the relevant research in the field. The presented results are reported in a clear and interpretable manner and give an overview the different recommendations. The subsumption and conclusion are conclusive and align with the cited references. There are only few minor points:

  1. Page 3, bottom paragraph: “screening and follow-up tool in U.S. and European studies.”
  2. The author should describe, how literature research was performed and what were the inclusion/exclusion criteria?

Author Response

There are only few minor points:

  1. Page 3, bottom paragraph: “screening and follow-up tool inS. and European studies.”

This typo was corrected

  1. The author should describe, how literature research was performed and what were the inclusion/exclusion criteria?

A “Methods” section has been added to page 2

Reviewer 3 Report

Comments and Suggestions for Authors

This article is a non systematic review of the evidence in melanoma, focusing on clinical and imaging follow-up and guidelines. The review is adequately structured.

Comments:

1. Guidelines:

1.1 Melanoma is an example of a tumor that requires multidisciplinary management. The European Cancer Organisation published guidelines for the management of these patients. I suggest including a phrase on the relevance of multidisciplinary management.

1.2. FDG PET/CT: This imaging technique is key for staging and restaging advanced melanoma. I recommend making a brief statement on the importance of performing it following international standardized procedures in order to make results comparable between different centers.

1.3. PET/MR: PET/MR is an imaging technique that is not available in many centers, although it is of great interest in melanoma, as it allows a precise evaluation of soft tissues and the brain, as it combines MR with PET. I suggest commenting this and referring to the EANM consensus recommendation for PET/MR

2. Meta-analyses related to melanoma. The following meta-analyses could be of interest for this review:

2.1 Meta-analysis of FDG PET/CT in melanoma: A meta-analysis of PET/CT in melanoma published in 2010, the result of a joint effort of Stanford and Madrid, could be of interest. The reference is: Jiménez-Requena F, Delgado-Bolton RC, Fernández-Pérez C, Gambhir SS, Schwimmer J, Pérez-Vázquez JM, Carreras-Delgado JL. Meta-analysis of the performance of (18)F-FDG PET in cutaneous melanoma. Eur J Nucl Med Mol Imaging. 2010 Feb;37(2):284-300. doi: 10.1007/s00259-009-1224-8. Epub 2009 Sep 2. PMID: 19727717; PMCID: PMC2886141.

3. Other comments:

3.1 Page 4, first paragraph: The following phrase "However, the use of PET-CT is limited by the delivery of higher energy positron radiation, expense, and availability, and recent years have seen dramatic improvements in the sensitivity of whole body CT. A 2024 study showed no difference in sensitivity and specificity between PET-CT and whole body CT when used to stage advanced cancer.(19)" and reference 19 corresponds to an article on ovarian cancer patients and not melanoma. This should be indicated in the text. It could be complemented with the following reflection: In the case of peritoneal implants, FDG PET/CT can show low FDG uptake; in this case it is important to analyse in detail the CT images and the areas of FDG uptake, to evaluate the possibility of peritoneal implants.

3.2 FDG/PET CT in melanoma patients treated with immunotherapy: These patients present particularities regarding the FDG PET findings and the interpretation. A recent review summarises the main issues that have to be taken into account: Mangas Losada M, Romero Robles L, Mendoza Melero A, García Megías I, Villanueva Torres A, Garrastachu Zumarán P, Boulvard Chollet X, Lopci E, Ramírez Lasanta R, Delgado Bolton RC. [18F]FDG PET/CT in the Evaluation of Melanoma Patients Treated with Immunotherapy. Diagnostics (Basel). 2023 Mar 4;13(5):978. doi: 10.3390/diagnostics13050978. PMID: 36900122; PMCID: PMC10000458.

3.3 Sentinel node biopsy in melanoma. Sentinel node biopsy is part of the standard of care in melanoma patients. It should be underlines that SPECT/CT can help improve the sentinel node biopsy and is recommended. 

Author Response

Guidelines:

 I suggest including a phrase on the relevance of multidisciplinary management.

Done-added to page 3

FDG PET/CT: This imaging technique is key for staging and restaging advanced melanoma. I recommend making a brief statement on the importance of performing it following international standardized procedures in order to make results comparable between different centers.

Done -added to page 4

PET/MR: PET/MR is an imaging technique that is not available in many centers, although it is of great interest in melanoma, as it allows a precise evaluation of soft tissues and the brain, as it combines MR with PET. I suggest commenting this and referring to the EANM consensus recommendation for PET/MR

The author appreciates this comment. Because PET/MR is not available in most centers it has not been included in guidelines, reviews, or large studies, and was therefore beyond the scope of this review.

Meta-analyses related to melanoma. The following meta-analyses could be of interest for this review:

  • Meta-analysis of FDG PET/CT in melanoma: A meta-analysis of PET/CT in melanoma published in 2010, the result of a joint effort of Stanford and Madrid, could be of interest. The reference is: Jiménez-Requena F, Delgado-Bolton RC, Fernández-Pérez C, Gambhir SS, Schwimmer J, PérezVázquez JM, Carreras-Delgado JL. Meta-analysis of the performance of (18)F-FDG PET in cutaneous melanoma. Eur J Nucl Med Mol Imaging. 2010 Feb;37(2):284-300. doi: 10.1007/s00259-009-1224-8. Epub 2009 Sep 2. PMID: 19727717; PMCID: PMC2886141.

The author agrees; this paper is cross -referenced in many of the included citations.

Page 4, first paragraph: The following phrase "However, the use of PET-CT is limited by the delivery of higher energy positron radiation, expense, and availability, and recent years have seen dramatic improvements in the sensitivity of whole body CT. A 2024 study showed no difference in sensitivity and specificity between PET-CT and whole body CT when used to stage advanced cancer.(19)" and reference 19 corresponds to an article on ovarian cancer patients and not melanoma. This should be indicated in the text. It could be complemented with the following reflection: In the case of peritoneal implants, FDG PET/CT can show low FDG uptake; in this case it is important to analyse in detail the CT images and the areas of FDG uptake, to evaluate the possibility of peritoneal implants.

The author agrees, and added two statements to page 4 clarifying these points.

FDG/PET CT in melanoma patients treated with immunotherapy: These patients present particularities regarding the FDG PET findings and the interpretation. A recent review summarises the main issues that have to be taken into account: Mangas Losada M, Romero Robles L, Mendoza Melero A, García Megías I, Villanueva Torres A, Garrastachu Zumarán P, Boulvard Chollet X, Lopci E, Ramírez Lasanta R, Delgado Bolton RC. [18F]FDG PET/CT in the Evaluation of Melanoma Patients

Treated       with      Immunotherapy.      Diagnostics      (Basel).      2023    Mar                4;13(5):978.            doi:

10.3390/diagnostics13050978. PMID: 36900122; PMCID: PMC10000458.

The author appreciates this reference and has added it to the text on page 10 and to the references.

Sentinel node biopsy in melanoma. Sentinel node biopsy is part of the standard of care in melanoma patients. It should be underlines that SPECT/CT can help improve the sentinel node biopsy and is recommended. The author whole heartedly agrees, However, the review is about follow-up imaging. Therefore SPECT CT was not relevant to it.